# A Mediation Approach to Discovering Causal Relationships between the Metabolome and DNA Methylation in Type 1 Diabetes

**DOI:** 10.3390/metabo11080542

**Published:** 2021-08-14

**Authors:** Tim Vigers, Lauren A. Vanderlinden, Randi K. Johnson, Patrick M. Carry, Ivana Yang, Brian C. DeFelice, Alexander M. Kaizer, Laura Pyle, Marian Rewers, Oliver Fiehn, Jill M. Norris, Katerina Kechris

**Affiliations:** 1Department of Biostatistics and Informatics, Colorado School of Public Health, University of Colorado Denver, Aurora, CO 80045, USA; alex.kaizer@cuanschutz.edu (A.M.K.); laura.pyle@ucdenver.edu (L.P.); katerina.kechris@cuanschutz.edu (K.K.); 2Barbara Davis Center for Diabetes, University of Colorado, Aurora, CO 80045, USA; patrick.carry@cuanschutz.edu (P.M.C.); jill.norris@cuanschutz.edu (J.M.N.); 3Department of Epidemiology, Colorado School of Public Health, University of Colorado Denver, Aurora, CO 80045, USA; lauren.vanderlinden@cuanschutz.edu (L.A.V.); marian.rewers@cuanschutz.edu (M.R.); 4Division of Biomedical Informatics and Personalized Medicine, School of Medicine, University of Colorado, Aurora, CO 80045, USA; randi.johnson@cuanschutz.edu (R.K.J.); ivana.yang@cuanschutz.edu (I.Y.); 5West Coast Metabolomics Center, University of California, Davis, CA 95616, USA; brian.defelice@czbiohub.org (B.C.D.); ofiehn@ucdavis.edu (O.F.)

**Keywords:** DNA methylation, metabolomics, type 1 diabetes

## Abstract

Environmental factors including viruses, diet, and the metabolome have been linked with the appearance of islet autoimmunity (IA) that precedes development of type 1 diabetes (T1D). We measured global DNA methylation (DNAm) and untargeted metabolomics prior to IA and at the time of seroconversion to IA in 92 IA cases and 91 controls from the Diabetes Autoimmunity Study in the Young (DAISY). Causal mediation models were used to identify seven DNAm probe-metabolite pairs in which the metabolite measured at IA mediated the protective effect of the DNAm probe measured prior to IA against IA risk. These pairs included five DNAm probes mediated by histidine (a metabolite known to affect T1D risk), one probe (cg01604946) mediated by phostidyl choline p-32:0 or o-32:1, and one probe (cg00390143) mediated by sphingomyelin d34:2. The top 100 DNAm probes were over-represented in six reactome pathways at the FDR <0.1 level (*q* = 0.071), including transport of small molecules and inositol phosphate metabolism. While the causal pathways in our mediation models require further investigation to better understand the biological mechanisms, we identified seven methylation sites that may improve our understanding of epigenetic protection against T1D as mediated by the metabolome.

## 1. Introduction

Type 1 diabetes (T1D) is an autoimmune disease characterized by the production of antibodies which target pancreatic β-cells. The disease currently affects over 30 million people worldwide [1] and is increasing by 3–4% per year on average [2].

Genetic predisposition accounts for some of the etiology of T1D (siblings of an individual with T1D have a relative risk 15 times higher than those without a sibling with T1D) [3] and explains some geographic variation in incidence [2]. Human leukocyte antigen (HLA) genes were the first to be linked to T1D and account for much of the genetic predisposition to the disease, but genome-wide association studies (GWAS) have also identified more than 40 other T1D-associated loci [4]. However, it is still unclear how these multiple loci [5] interact with one another and the environment to produce a T1D diagnosis. In addition to the complex genetics of T1D, low monozygotic (MZ) twin concordance (approximately 50%) and the increasing incidence over time [2] support the theory that non-genetic factors play an important role in T1D development [1].

Epigenetic differences may be important contributors to T1D etiology. Changes in methylation have been associated with other autoimmune conditions, and monozygotic twins can be epigenetically heterogeneous despite sharing an identical genetic code [1]. Rakyan et al. [1] and Stefan et al. [6] performed epigenome-wide association studies (EWAS) in discordant and concordant twin pairs and found that methylation profiles were more similar among participants with T1D than in unaffected twins. Epigenetics profiles were also combined with GWAS data and differentially methylated sites were mapped to six well-known T1D susceptibility genes, including two major histocompatibility complex (MHC) genes and several HLA loci [1,6]. Finally, Johnson et al. found that T1D cases had different longitudinal methylation patterns compared to controls prior to diagnosis [7].

Environmental factors including viruses, diet, and the metabolome have also been linked with islet autoimmunity (IA) [8] and T1D etiology [4]. Previous studies have found associations between T1D and differentially expressed phospholipids and sphingolipids, excretion of modified amino acids, and vitamin D (and related compounds on its metabolic pathway) [4,9].

To date, the effects of the metabolome and DNA methylation have been studied separately in T1D; thus, the combined nutrigenomics of T1D remain unclear. Early in vitro studies confirmed metabolome-dependent alterations in DNA methylation associated with various cancers [10,11], and even identified oncometabolites associated with the development of glioma [12]. More recent human studies also support the connection between the metabolome and methylation in breast cancer [13], colorectal cancer [14], and smoking-related diseases [15]. A 2018 study by Zaghlool et al. used linear models to link metabolomics and DNA methylation with type 2 diabetes (T2D) and obesity and used Mendelian randomization (MR) to provide evidence for “a causal effect of metabolite levels on methylation of obesity-associated CpG sites” [16].

Causal interpretations are of particular interest when integrating epigenetic and metabolomics data. The counterfactual framework for mediation analyses is widely used in statistics and epidemiology because it can allow for causal interpretations of observational data. Additionally, it provides a relatively intuitive interpretation of mediation effects by quantifying how much an outcome of interest would likely differ given a change in either the exposure or mediator [17]. Because mediation is defined based on an assumed causal model, it is important that causality assumptions are reasonable [18]. It is difficult to evaluate causal assumptions across the epigenome, but longitudinal study designs allow for mediation models in which the exposure variable is measured prior to the mediator. To our knowledge, this is the first study to examine causal links between DNAm, the metabolome, and T1D.

The Diabetes Autoimmunity Study in the Young (DAISY) follows 2547 high-risk children in Colorado for the development of IA and T1D (ClinicalTrials.gov identifier: NCT03205865). Follow-up includes blood sample collection at 9, 15, and 24 months, then annual collection until one or more autoantibodies are detected. Participants who test positive for one or more autoantibodies are asked to follow an accelerated protocol with visits and blood sample collection every 3–6 months. IA is defined as two consecutive visits at which a confirmed autoantibody to insulin, GAD65, IA-2, or ZnT8, was detected (Figure 1) [19]. The children are followed over time, until they are diagnosed with T1D by a physician using American Diabetes Association criteria. Thus, DAISY’s study design allows us to include exposure and mediator variables that are temporally separated in our mediation models because methylation was measured prior to seroconversion (PSV), defined as the autoantibody-negative visit preceding the visit at which the child first tested autoantibody positive (SV), and metabolite was measured at SV and vice versa. This aids in interpretation of the results and in theory does not violate the assumptions of a mediation analysis.

## 2. Results

A total of 183 participants had both metabolite and methylation data at their PSV and SV study visits. Figure 1 outlines DAISY study visits. Participant characteristics are shown in Table 1, and there were no statistically significant differences between IA cases and controls with respect to age, sex, race/ethnicity, DR3/4 status, and first-degree relative (FDR) status.

Out of a total 379,557,915 possible pairs, our linear regression filtering step identified 1490 pairs with methylation measured at PSV and metabolite measured at SV where metabolite and methylation probe were significantly associated with one another, and both were associated with IA at the *p* < 0.01 level (Figure 1b). We also identified 600 pairs with metabolite measured at PSV and methylation measured at SV for mediation analysis (Figure 1c). Correlations between DNAm and metabolite for all candidates are shown in Figure 2.

After FCR adjustment, seven pairs had a confidence interval for the estimate of NIE that did not contain 0 on the log odds scale (Table 2). Each pair contained a unique probe, but the metabolite histidine was the mediator in five of the pairs. No pairs in which metabolite was measured at PSV and methylation was measured at SV were significant after FCR adjustment. All confidence intervals are presented after FCR adjustment. For all seven pairs, the metabolite was positively associated with IA, while methylation was negatively associated with both the outcome and respective metabolite. Thus, an indirect effect less than one suggests that some of the protective effect of methylation is via reduction in the metabolite levels.

Of the unique probes, six measure methylation at a site in a known gene, and four genes are protein coding. Of the four probes linked to protein-coding genes, two are located within a CpG dinucleotide island (cg00390143 and cg01172082), one on the southern shore of the CpG island (cg07964219), and one in open sea (cg01604946). These probes are associated with the GALNT9, KIF26A, COL18A1, and SH3TC2 genes, respectively. The probe associated with HLA-DQB2 (cg19939773) was also located within a CpG island, while cg15052330 (linked to the CYP26B1 gene) was in open sea and cg15688253 was on the north shore of a CpG island. The metabolite histidine mediated the effect of the five methylation sites associated with the CYP26B1, HLA-DQB2, KIF26A, and COL18A1 genes, while a phosphatidyl choline (PC) identified as PC (p-32:0) or PC (o-32:1) mediated the effect of cg01604946 (located in the SH3TC2 gene), and a sphingomyelin (SM) we identified as SM (d34:2) mediated the effect of cg00390143 (GALNT9 gene).

Six reactome pathways were enriched (FDR *q*-value < 0.10) (Table 3), including plasma lipoprotein assembly, remodeling, and clearance and inositol phosphate metabolism. All pathways were enriched at least two-fold. Over-representation of the inositol phosphate metabolism pathway was driven by the INPP5A and PLCH2 genes (associated with probes cg13931663 and cg20468586, respectively, from our list of the top 100 probes). The plasma lipoprotein assembly, remodeling, and clearance pathway over-representation was driven by the LIPA and PCSK6 genes (probes cg24405248 and cg07375207, respectively).

None of the biopathways were significant after *p*-value adjustment (Table 4). However, several immune-system-related pathways were significant at an unadjusted *p* < 0.05 level, including interleukin signaling and transmembrane transport of small molecules. Interestingly, transport and metabolism of small molecules appeared in both the over-representation and enrichment analyses.

## 3. Discussion

We found seven unique pairs of CpG dinucleotide sites and metabolites with significant natural indirect mediation effects. These pairs included five DNAm probes mediated by histidine (a metabolite known to affect T1D risk), one probe (cg01604946) mediated by phostidyl choline p-32:0 or o-32:1, and one probe (cg00390143) mediated by sphingomyelin d34:2. For all seven pairs, the metabolite was positively associated with IA, and the methylation of the CpG site was negatively associated with both the metabolite and IA. Thus, an indirect effect less than one suggests that some of the protective effect of the CpG site is via reduction in metabolite levels.

We previously discovered nominally significant positive associations between histidine and the risk of progression from IA to T1D [20], and other studies indicated that the histidine–glutamate–glutamine pathway can be corrected by improving glycemic control [21]. Additionally, histidine is a precursor to histamine, a monoamine that plays an important role in inflammation and has been linked to the development of T1D and other immune responses [22,23]. A knock-out mouse experiment for the gene encoding histidine decarboxylase (the enzyme that converts histidine to histamine) showed that decreased histamine levels were associated with a lower incidence of T1D and decreased levels of circulating IL-12 and IFN-γ [22].

While the connection between histamine and T1D is relatively well studied, it remains unclear how it might mediate the protective effects of the CpG sites we identified, and how the methylation of those CpG sites might affect T1D etiology. The association of IA and the CpG site profiled by probe cg19939773, located approximately 1.4 kb upstream of the second exon of the HLA-DQB2 gene, was mediated by histidine. HLA-DQB1 is an important risk gene in development of T1D, but the literature on HLA-DQB2 is less clear [24]. Interestingly, methylation of the KIF26A gene (probe cg01172082), which was also mediated by histidine, was found to be affected by a 5-day high-fat high-energy diet in young men, and these changes could potentially contribute to the insulin resistance seen in some of the study participants with T2D [25].

Das et al. found using whole-exome sequencing (WES) that the *COL18A* gene (which we found was mediated by histidine) is protective against diabetic retinopathy, but the sample size was small, and these results warrant replication [26]. A relatively large study also found that polymorphisms in the collagen protein encoded by COL18A may be related to increased obesity in patients with T2D [27]. COL18A antisense RNA is also not well understood, but mouse models have provided some evidence that other antisense RNAs, namely GLUT-2, can increase risk of diabetes, [28] so it is possible that increased methylation results in lower expression of risk-increasing antisense RNA.

For the phosphatidyl choline (PC) identified as either PC (p-32:0) or PC (o-32:1) and the sphingomyelin (SM) identified as SM (d34:2), a study by Oresic et al. found that participants who went on to develop T1D had lower levels of PCs at birth, but that “the lysoPC PC(18:0/0:0) was increased 1.5-fold within 9–18 mo before seroconversion” [8]. Elevated lysoPCs are believed to be a marker of oxidative stress prior to the development of islet autoantibodies [29], but our group also found that PC (16:0_18:1(9Z)) was the strongest single metabolite predictor of IA reversion. The CpG site profiled by the cg01604946 probe is located within the SH3TC2 gene and its association with IA is mediated by the compound identified as PC (p-32:0) or PC (o-32:1). This phosphatidyl choline is generally associated with demyelinating motor and sensory neuropathy, and its connection to T1D requires more research [30].

Like PCs, the research regarding SMs generally suggests that they have a protective effect against developing T1D [8,31] and mouse models have shown that SM patches on pancreatic β-cells correlate well with insulin production capacity [32]. However, SMs are also strongly associated with rapid eGFR decline [33] and general nephropathy [34] in those who already have T1D. Both SMs and PCs have also been linked to renal impairment and all-cause mortality in T1D, [35] which indicates that understanding of their role in T1D etiology remains incomplete and specific SMs and PCs may not be protective. Like the COL18A and KIF26A genes, which were mediated by histidine, the CpG probed by cg00390143 was associated with the GALNT9 gene, which has been implicated in T2D but remains understudied [36].

This study has identified several potential causal pathways in the etiology of IA. The seven methylation sites that were significantly mediated by metabolites are potentially interesting candidates for elucidating epigenetic protection against T1D. While the causal pathways in our mediation models are temporally reasonable in the sense that the exposure was measured prior to the mediator, a better understanding of the biological pathways is necessary to confirm these exploratory analyses.

A limitation of this work is that metabolites were measured in non-fasting samples, and these analyses did not account for dietary intake, which is the single biggest source of exposure to chemicals and nutrients [37]. Although DAISY has collected dietary information, these data are available on only a subset of the relevant timepoints herein; therefore, adjustment for dietary intake would have significantly decreased our sample size. However, case/control status for each participant is unknown at the time of metabolite measurement, so participants are effectively randomized with respect to outcome. Thus, we would not expect there to be a systematic difference in diet that would affect these results.

Enrichment and over-representation tools are not designed for the integration of methylation and metabolomics data. To our knowledge, there are no integrated enrichment or over-representation tools that can incorporate multiple omics datasets, so our biological interpretation relied on an ad hoc choice of the 100 most significant DNAm probes.

Additionally, we did not adjust the candidate selection step for multiple testing, and only adjusted *p* values at the second stage, which could potentially lead to false positives. However, the best approach to p-value adjustment for correlated variables in this filtering step remains an active area of statistical research and we suggest that this approach is unlikely to result in many false positives due to the adjustments used in the later stages [38].

However, despite these limitations we believe that our approach is a novel method for the integration of omics data in epidemiological studies. The combination of a longitudinal study design and mediation analysis allows for causal interpretation of our results, which will hopefully guide additional research into biological mechanisms. Additionally, this approach is not limited to DNAm and metabolite studies and could in theory be applied to any longitudinal study with multiple omics datasets.

## 4. Materials and Methods

### 4.1. Study Design and Participants

Study participants were recruited via newborn screening at St. Joseph’s Hospital in Denver, CO, USA and from unaffected first-degree relatives (FDR) of type 1 diabetes patients. For the DAISY nested case–control study, IA cases were frequency-matched to controls by age at SV, race/ethnicity, and sample availability. The majority of participants were Non-Hispanic White (NHW), and race/ethnicity was categorized into NHW and Other for matching and analysis. All research was performed in accordance with relevant guidelines and regulations [7,39,40]. Participants with both methylation and metabolomic measures at the visit at PSV and SV were selected for these analyses (*n* = 183).

### 4.2. DNA Methylation

IA cases and frequency-matched controls were randomly assigned to either the 450 K group (which included duplicate samples for quality control) or EPIC group (which included replicates from the 450 K set for quality control). DNA methylation was profiled in peripheral whole blood using the Infinium HumanMethylation450K Beadchip (Illumina, San Diego, CA, USA, “450 K”) for the 450 K set, and the Infinium HumanMethylation EPIC Beadchip (“EPIC”) was used for the EPIC set. Both sets underwent identical pre-processing using the SeSAMe pipeline [41], and measurement platform was included as a covariate in all statistical models in order to account for technological batch effects. Johnson and colleagues performed quality control and removed poor-quality samples and DNAm probes (see Johnson et al. [7] for details). This resulted in 199,243 quality DNA methylation probes and 183 subjects having DNAm at both PSV and SV timepoints. Normalized M values were used in all statistical analyses. We use the term DNAm probe and the probe identifier when referring to the data in the results. However, each probe is designed to measure methylation at a CpG site which is used as a more general term in the discussion.

### 4.3. Metabolomics

Untargeted metabolomics were obtained using gas chromatography–time-of-flight mass spectrometry (GC–TOF MS), charged surface hybrid column quadrupole time-of-flight mass spectrometry (CSH–QTOF MS), and hydrophilic interaction chromatography quadrupole time-of-flight mass spectrometry (HILIC–QTOF MS) at the UC Davis West Coast Metabolomics Center. Non-fasting plasma samples were prepared and analyzed as previously described in Johnson et al. [20].

For GC–TOF data, peak picking and annotation was performed using BinBase [42]. CSH–QTOF MS and HILIC–QTOF MS were processed using MS-Dial [43], complex lipids were annotated with LipidBlast [44] and Massbank of North America (http://mona.fiehnlab.ucdavis.edu/, accessed on 6 August 2021), and erroneous peaks were removed using MSFLO [45].

After collection, annotation, and post-processing, metabolomics data were normalized using SERRF [46], a QC-based method designed to account for batch effects. Samples with low abundance (*n* = 2) and metabolites with a coefficient of variation more than two absolute deviations from the median (*n* = 344) were excluded, and data were transformed using the Box-Cox method [47]. After processing and quality checks, 2457 untargeted metabolites remained, of which 1905 metabolites were annotated with either a lab-specific or InChIKey identifier [20]. Only annotated metabolites were considered for these analyses.

### 4.4. Statistical Analysis

All analyses were performed using the R programming environment [48] version 4.0.0 and included only participants with methylation and metabolomics data at both their SV and PSV study visits (*n* = 183). To reduce the number of methylation probe–metabolite pairs examined in the mediation analysis, we performed simple linear regression to find probes and metabolites that were correlated at a nominal *p* < 0.01 level. Next, we performed two independent logistic regressions on all significantly correlated pairs (one model for DNAm probe and one for metabolite) to determine pairs in which both were significantly associated with the IA outcome, again at a nominal 0.01 level. Pairs in which both metabolite and probe were correlated with one another, and both variables were associated with the outcome, were selected for the mediation analysis. These *p* values were not adjusted for multiple comparisons because this was a candidate-filtering step intended to save computing time for later steps.

Model-based mediation analyses were performed using the regmedint package [49] version 0.2.1, an R implementation of Valeri and VanderWeele’s SAS macros [50]. We report the natural indirect effect (NIE). The natural indirect effect is interpreted on the odds scale and represents the average change in outcome if the exposure a (methylation at PSV) was fixed at 1, but the mediator t (metabolite at SV) were changed from the level it would take if a=0 to the level it would take if a=1 [17]. In other words, it is the effect of the exposure on the outcome that operates through changing the mediator [17].

All regression models were adjusted for HLA-DR3/4 haplotype, age at PSV, time from PSV to SV, and sex, and included exposure mediator interaction terms as recommended by VanderWeele [17]. Confidence intervals for the NIE were calculated based on 10,000 bootstrap simulations using the percentile method and adjusted using Benjamini and Yekutieli’s False Coverage–Statement Rate (FCR) method [51] at the *q* = 0.05 level (equivalent selection based on a false discovery rate-adjusted *p* value < 0.05). Because DAISY is a case–control study, cases are oversampled relative to the general population. To account for this, the regmedint package fits the mediator model using only the control participants, which approximates the results from a cohort study when the outcome is rare [17].

### 4.5. Biological Interpretation

Pathway over-representation was performed using PANTHER’s [52] implementation of reactome pathways [53] and included 60 unique genes associated with the top 100 probes from the mediation step of the analysis. These *p* values were based on the standard error of the NIE estimated using the multivariate delta method, as opposed to being converted from bootstrap confidence intervals [17]. CpG sites were converted to Entrez IDs using the missMethyl R package [54] and the 199,243 original probes were used as the reference list. Pathways with less than 5 genes in the reference list and less than two genes in the analysis list were excluded.

Pathway enrichment analysis was performed using the Enrichr (https://maayanlab.cloud/Enrichr/, accessed on 6 August 2021) [55] Biopathways 2019 module [56], again with the 60 unique genes associated with the top 100 candidate probes.

We also searched for known genotypes at certain loci with allele-specific methylation using the BIOS QTL browser (https://genenetwork.nl/biosqtlbrowser/, accessed on 6 August 2021) [57]. These methylation quantitative trait loci (meQTLs) can alter methylation patterns across genomic regions [58]. We then used the biomaRt R package [59] to obtain gene symbols, which were annotated using GeneCards [60] via the GeneBook R package [61].

## Figures and Tables

**Figure 1 metabolites-11-00542-f001:**
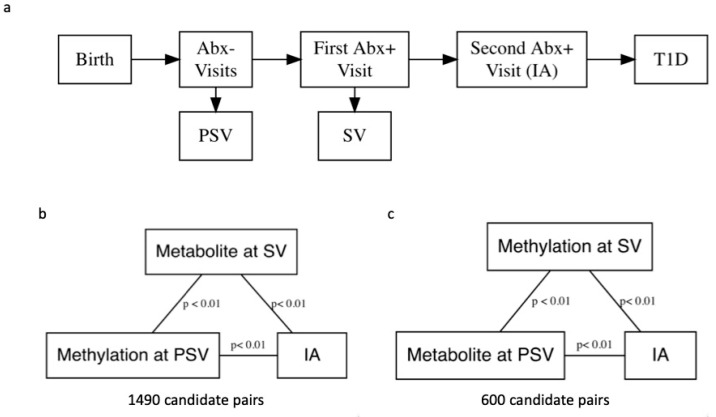
(**a**) A flowchart depicting progression from birth to islet autoimmunity (IA) and the Diabetes Autoimmunity Study in the Young (DAISY) visits. To reduce computation time, we performed simple linear regression without covariate adjustment to identify pairs of DNA methylation (DNAm) probes and metabolites for mediation analysis. Candidate pairs were significantly associated at the *p* < 0.01 level, and both DNAm probe and metabolite were associated with development of IA at the *p* < 0.01 level. (**b**) We examined pairs with DNAm measured pre-seroconversion (PSV) and metabolite measured at seroconversion (SV). (**c**) We also examined pairs with metabolite measured at PSV and DNAm measured at SV in separate analyses.

**Figure 2 metabolites-11-00542-f002:**
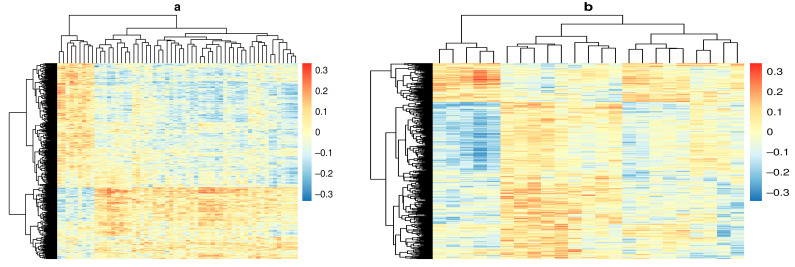
Heat maps depicting the correlation coefficient between DNA methylation (DNAm) probes in rows and metabolites in columns. Columns and rows were clustered using the complete linkage method. (**a**) Correlation between DNAm at pre-seroconversion (PSV) and metabolite measured at seroconversion (SV). (**b**) Correlation between DNAm at seroconversion (SV) and metabolite measured pre-seroconversion (PSV).

**Table 1 metabolites-11-00542-t001:** Participant characteristics at first visit.

	Case (*n* = 92)	Control (*n* = 91)	Total (*n* = 183)	*p* Value
**Age**				0.387 ^1^
Mean (SD)	6.2 (4.3)	6.8 (4.2)	6.5 (4.3)	
Range	0.7–18.3	0.7–20.3	0.7–20.3	
**Non-Hispanic White**				0.756 ^2^
No	22 (23.9%)	20 (22.0%)	42 (23.0%)	
Yes	70 (76.1%)	71 (78.0%)	141 (77.0%)	
**DR3/4**				0.172 ^2^
No	67 (72.8%)	74 (81.3%)	141 (77.0%)	
Yes	25 (27.2%)	17 (18.7%)	42 (23.0%)	
**Sex**				0.599 ^2^
Female	44 (47.8%)	40 (44.0%)	84 (45.9%)	
Male	48 (52.2%)	51 (56.0%)	99 (54.1%)	
**FDR Status**				0.819 ^2^
First-degree relative with T1D	49 (53.3%)	50 (54.9%)	99 (54.1%)	
General population (no first degree relative with T1D)	43 (46.7%)	41 (45.1%)	84 (45.9%)	

^1^ Linear model ANOVA; ^2^ Pearson’s Chi-squared test.

**Table 2 metabolites-11-00542-t002:** Methylation–metabolite pairs with significant natural indirect effects.

Metabolite	Probe	Position	Relation to Island	Gene	Type	Description	NIE	NIE CI Low	NIE CI High
Histidine	cg15688253	chr1:1096717	N_Shore				0.766	0.418	0.993
Histidine	cg15052330	chr2:72360243	OpenSea	CYP26B1			0.714	0.339	0.950
PC (p-32:0) or PC (o-32:1)	cg01604946	chr5:148398804	OpenSea	SH3TC2	Protein Coding	SH3 Domain; Tetratricopeptide Repeats 2; SH3TC2 Divergent Transcript	0.663	0.342	0.895
Histidine	cg19939773	chr6:32729876	Island	HLA-DQB2			0.785	0.439	0.994
SM (d34:2) [M+HAc-H]- & [M+Cl]- _YLWSJLLZUHSIEA-CKSUKHGVSA-N	cg00390143	chr12:132842539	Island	GALNT9	Protein Coding	Polypeptide N-Acetylgalactosaminyltransferase 9	0.777	0.494	0.999
Histidine	cg01172082	chr14:104645732	Island	KIF26A	Protein Coding	Kinesin Family Member 26A	0.759	0.390	0.989
Histidine	cg07964219	chr21:46847898	S_Shore	COL18A1	Protein Coding	Collagen Type XVIII Alpha 1 Chain; COL18A1 Antisense RNA 1; COL18A1 Antisense RNA 2	0.795	0.447	0.986

**Table 3 metabolites-11-00542-t003:** Reactome pathway enrichment.

Term	Number of Genes in Top 100 Probes	Number of Genes in Reference List	Expected	Fold Enrichment	*p* Value	FDR *q* Value
Formation of the cornified envelope (R-HSA-6809371)	3	105	0.32	9.24	0.004	0.071
Inositol phosphate metabolism (R-HSA-1483249)	2	43	0.13	15.04	0.008	0.071
Ion transport by P-type ATPases (R-HSA-936837)	2	49	0.15	13.2	0.011	0.071
Keratinization (R-HSA-6805567)	3	159	0.49	6.1	0.014	0.071
Transport of small molecules (R-HSA-382551)	6	668	2.07	2.91	0.017	0.071
Plasma lipoprotein assembly, remodeling, and clearance (R-HSA-174824)	2	63	0.19	10.27	0.017	0.071

**Table 4 metabolites-11-00542-t004:** Bioplanet enrichment (top ten).

Term	Number of Genes in Top 100 Probes	Number of Genes in Reference List	*p* Value	FDR *q* Value	Fold Enrichment	Genes
Interleukin receptor SHC signaling	2	28	0.0031816	0.2497991	26.411141	IL3;IL5RA
Ion transport by P-type ATPases	2	36	0.0052218	0.2497991	20.188641	ATP4B;ATP2B2
Interleukin-3 signaling pathway	2	45	0.0080651	0.2497991	15.955894	IL3;IL5RA
Interleukin-3, interleukin-5, and GM-CSF signaling	2	45	0.0080651	0.2497991	15.955894	IL3;IL5RA
Regulation of NFAT transcription factors	2	47	0.0087727	0.2497991	15.245211	IL3;IKZF1
Ion channel transport	2	61	0.0144583	0.2497991	11.619521	ATP4B;ATP2B2
Sodium-coupled sulphate, di- and tri-carboxylate transporters	1	5	0.0149116	0.2497991	84.474576	SLC13A2
Cytochrome P450 metabolism of vitamins	1	6	0.0178676	0.2497991	67.576271	CYP26B1
Phase I of biological oxidations: non-cytochrome P450 enzymes	1	7	0.0208149	0.2497991	56.310735	LIPA
Eosinophils in the chemokine network of allergy	1	8	0.0237535	0.2497991	48.263922	IL3
Small cell lung cancer	2	84	0.0263601	0.2497991	8.350715	LAMA2;TRAF5
Hematopoietic cell lineage	2	88	0.0287269	0.2497991	7.960706	IL3;IL5RA
Transmembrane transport of small molecules	4	432	0.0405993	0.2497991	3.256342	ATP4B;SLC13A2;STEAP3;ATP2B2

## Data Availability

The data presented in this study are available on request from the corresponding author. The data are not publicly available because they contain protected health information.

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
