# Peer review of "A Mediation Approach to Discovering Causal Relationships between the Metabolome and DNA Methylation in Type 1 Diabetes"

_metabolites, 2021, doi:10.3390/metabo11080542_

Round 1
Reviewer 1 Report
Dear authors,
I found your paper interesting. However, I believe that it can be improved.
As minor chages, you must review the abbreviations used throughout the manuscript - some are present before presenting them, other are not used when they should, etc (rows 94, 159, 178-179, 188). Also, please check Figure 2 and insert it properly. Further, double-check all figures' and tables' captions.
As major concerns, I feel the results section is expediently presented. Perhaps more details/explanations would benefit the reader, although the discussion section is well-written and researched.
Indeed, the limitations you urderlined are a major concern, especially not accounting for dietary intake. Perhaps some food questionnaires/survey in order to assess that the two groups' nutritional intake doesn't differ significantly? Or some additional info from previously published data?
Please provide some details of the DAISY study or provide the link adress (clinicaltrials.gov?).
Author Response
- As minor chages, you must review the abbreviations used throughout the manuscript - some are present before presenting them, other are not used when they should, etc (rows 94, 159, 178-179, 188). Also, please check Figure 2 and insert it properly. Further, double-check all figures' and tables' captions.
Thank you for bringing these to our attention. We have clarified that CpG refers to a dinucleotide methylation site and believe that the abbreviations are now consistent. Also, the Metabolites editorial team kindly assisted with formatting tables and figures, so we will work with them to make sure figures are properly inserted for publication.
- As major concerns, I feel the results section is expediently presented. Perhaps more details/explanations would benefit the reader, although the discussion section is well-written and researched.
Thank you for this comment. We have added additional details about the relationships between methylation probe and metabolite and the genes driving enrichment and overrepresentation to the results section. We agree that this has significantly improved the manuscript.
- Indeed, the limitations you urderlined are a major concern, especially not accounting for dietary intake. Perhaps some food questionnaires/survey in order to assess that the two groups' nutritional intake doesn't differ significantly? Or some additional info from previously published data?
Unfortunately, an analysis of dietary intake was not possible for this analysis, because only about 75% of the participants had diet data at the relevant timepoints and limiting the analysis to these participants would have greatly reduced our power. However, case/control status for each participant is unknown at the time of metabolite measurement, so participants are effectively randomized with respect to outcome. We have added this information to the manuscript and hope it will put readers’ minds at ease.
- Please provide some details of the DAISY study or provide the link adress (clinicaltrials.gov?).
We have added DAISY’s clinicaltrials.gov identifier on line 84.

Reviewer 2 Report
The main premise of this paper is very interesting and focusing on combined analyses of metablomics and methylation which are the key elements in the development of many diseases and Type 1 Diabetes which is the focus of this study. Though sample size is small, it takes a target based approach which is useful in addressing the main questions posed by the authors. The topic is very novel and combined analyses have not been carried out previously with a mediation approach. To me the details are clear and the overall paper is clearly written. The conclusions are based on the obtained results and are well crafted and explained. The authors do address all their main questions to explain their observed associations.Author Response
The main premise of this paper is very interesting and focusing on combined analyses of metablomics and methylation which are the key elements in the development of many diseases and Type 1 Diabetes which is the focus of this study. Though sample size is small, it takes a target based approach which is useful in addressing the main questions posed by the authors. The topic is very novel and combined analyses have not been carried out previously with a mediation approach. To me the details are clear and the overall paper is clearly written. The conclusions are based on the obtained results and are well crafted and explained. The authors do address all their main questions to explain their observed associations.
We thank the reviewer for these positive comments.
Round 2
Reviewer 1 Report
The manuscript has improved.